# A Hybrid Optimization Approach for the Enhancement of Efficiency of a Piezoelectric Energy Harvesting System

**Mahidur R. Sarker** [1,2,*], **Ramizi Mohamed** [3], **Mohamad Hanif Md Saad** [1], **Muhammad Tahir** [4], **Aini Hussain** [3] and **Azah Mohamed** [3]

1   Institute of IR 4.0, Universiti Kebangsaan Malaysia, Bangi 43600, Malaysia; hanifsaad@ukm.edu.my
2   Industrial Engineering and Automotive, Nebrija University, Campus de la Dehesa de la Villa, Calle Pirineos, 55, 28040 Madrid, Spain
3   Department of Electrical, Electronic and Systems Engineering, Faculty of Engineering and Built Environment, Universiti Kebangsaan Malaysia, Bangi 43600, Malaysia; ramizi@ukm.edu.my (R.M.); draini@ukm.edu.my (A.H.); profazah@gmail.com (A.M.)
4   Department of Physics, Faculty of Physical and Numerical Sciences, Abdul Wali Khan University, Mardan 23200, Pakistan; tahir@awkum.edu.pk
*   Correspondence: mahidursarker@ukm.edu.my

**Abstract:** This paper presents a hybrid optimization approach for the enhancement of performance of a piezoelectric energy harvesting system (PEHS). The existing PEHS shows substantial power loss during hardware implementation. To overcome the problem, this study proposes a hybrid optimization technique to improve the PEHS efficiency. In addition, the converter design as well as controller technique are enhanced and simulated in a MATLAB/Simulink platform. The controller technique of the proposed structure is connected to the converter prototype through the dSPACE DS1104 board (dSPACE, Paderborn, Germany). To enhance the proportional-integral voltage controller (PIVC) based on hybrid optimization method, a massive enhancement in reducing the output error is done in terms of power efficiency, power loss, rising time and settling time. The results show that the overall PEHS converter efficiency is about 85% based on the simulation and experimental implementations.

**Keywords:** controller; hybrid optimization; energy harvesting; converter; piezoelectric vibration transducer

## 1. Introduction

Over the years, energy harvesters and converters have become an essential part of any energy harvesting system using wasted ambient energy to provide power for the anticipated future growing energy demand. Examples of wasted ambient energy sources are wind energy, thermal energy, sound energy, vibration energy, solid waste energy and solar energy [1–3]. The utilization of piezoelectric components to collect energy from encompassing vibrations is one of the essential renewable energy sources, mainly for remote areas that lack power [4,5]. On the other hand, due to the lack of solar energy, light efficiency might drop dramatically during overcast days, and a comparatively enormous surface region is needed relying upon the power prerequisites of the related electronic framework. Besides, a thermal energy source needs huge temperature differences to generate sufficient amounts of electrical energy. However, in a positive way this pollution-free and maintenance-free energy source has a long operating lifetime.

Energy harvesting (EH) through piezoelectric elements falls under the micro scale category. The EH through piezoelectricity utilizes the direct piezoelectric effect. The characteristics of piezoelectric element direct and converse effect are explained by Equations (1) and (2), respectively [6].

$$D = d\sigma + \varepsilon^\sigma E \tag{1}$$

$$S = S^E \sigma + dE \tag{2}$$

where $D$ is the polarization, $\sigma$ is the stress, $E$ is the electric field, $S^E$ is the elastic compliance, $d$ is the piezoelectric charge coefficient, $\varepsilon^\sigma$ is the absolute permittivity under constant stress, $S$ is the strain. The piezoelectric based EH can produce huge charge with suitable combination of Equations (1) and (2). Piezoelectric EH is one of the famous vibrational-based EH methods, the key feature of which is the piezoelectric property where this material produces electric charge by the application of mechanical stresses. On the other hand, application of electric potential, mechanical-deformation in the structure of the piezoelectric material is produced. The phenomenon will be clear as shown in Figure 1.

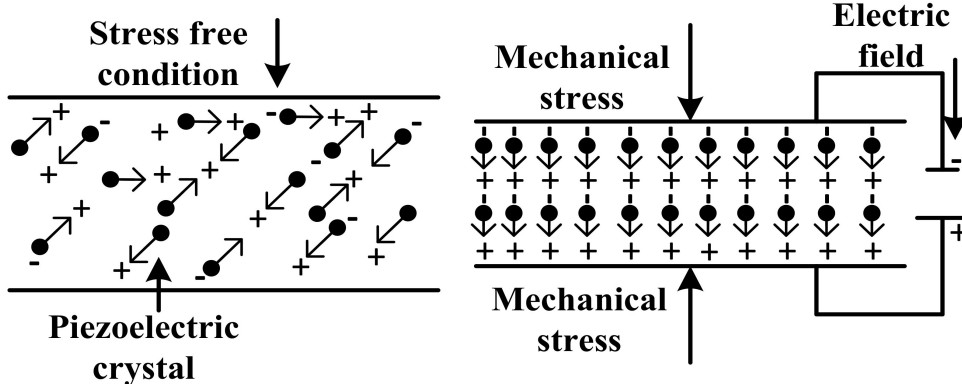

**Figure 1.** Piezoelectric materials producing an electrical charge with the application of mechanical stress.

Cantilever beams are the most used structures for micro EH; the key feature of their design is to operate at the resonance frequency [7]. In these (unimorph/bimorph) structures the dimensions are reduced, enabling the material to withstand higher stress levels. Unimorph EH is the simplest cantilever EH structure, containing a single layer of piezoelectric joined with another layer of non-piezoelectric material. In cantilever-structures this second layer is known as substrate-layer. In micro PEHS the: (a) high performance and (b) low-cost totally depend upon the new technologies. The basic requirements of these micro systems are: (a) ease of integration; (b) compactness and (c) being maintenance free [8]. Nowadays the main goal for researchers is the optimization of structural parameters i.e., (a) geometry; (b) length; (c) width and (d) thickness, of PEHS to achieve maximum power production. The previously famous D-shaped PEHS in cantilever-beam structures need to be replaced due to their poor average strain. In the last decades, researchers were more interested in piezoelectric materials and operating-modes, while the geometry of the cantilever-harvester was largely ignored. Now it is proved by studies that geometry is the most important parameter that affects the output-power of the harvester and this can be easily understand by the features [9]. When the geometry of the beam is changed the strain distribution along the length of the beam is varied, hence the power can be increased.

A PEHS based on vibration generates low AC power ($\mu$W-mW), thus a boost converter is required to make this output suitable for micro-power applications [10,11]. The improvement in efficiency of PEHS is closely related to the design of converter and controllers. Concerning the reason, power-losses becomes major attributes which relate to the efficiency of the converter that usually occurs during prototype implementation [12–14]. A proper switching frequency choice for the power accessory is essential when determining the most suitable PEHS converter model [15,16]. The selection of an appropriate switching frequency and components, which affect the converter output quality, power loss and efficiency, is another important issue in order to design a low power converter for micro-device applications [17–19]. In the vibration-based PEHS, the problem of an infrequent power environment in the ambient source should be considered in the converter model [20–22]. To extract energy from vibrations is a difficult task for several reasons, such as the unpredictable behavior of vibrations, spillage of precession of miniature energy collectors, driving and handling the extracted signal, as well as applying the collected miniature capacity to stack a suitable voltage and current.

Several controllers are devoted in the converter, namely, analogue circuit controllers, digital circuit controllers, artificial intelligent (AI) controllers, proportional integral derivative (PID) controllers, microcontrollers and fuzzy logic control (FLC). Typically, the PID controller is used to control different applications because of its usual controlling ability, simple turn, toughness and roughness [23–25]. Several advantages of using the PID controller are that it can solve numerous problems such as huge stable mistakes and rotations because of shifts in automated loads [26]. In [26], a traditional proportional-integral voltage controller (PIVC) is addressed to regulate a step-up converter to suite dynamic load variations. However, the drawback of a PIVC is its mathematical model and trial-and-error process [27,28]. The intricate aspect of a PIVC is to get the appropriate parameter values, namely a proportional gain ($Kp$) and an integral gain ($Ki$). Typically, to tune the suitable values of the PIVC manually is tedious [29]. However, this study has used a hybrid optimization technique to develop the conduct and determine the best parameter values of the PIVC. The lightning search algorithm (LSA) is an optimization method that has the advantage of stable durability and globe concurrence efficiency as well as being easy to implement [30]. The dSPACE DS1104 controller is a user-friendly tool used in MATLAB/Simulink for improving control methods and simulations. The actual model is created, loaded, and begins mechanically in the actual-time experimental, hence, reducing the complete structure process time [31]. This study has used the PIVC to create pulse width modulation (PWM) signals for metal-oxide semiconductor field effect transistors (MOSFETs) switching, so as to generate and control the DC harvest voltage. Therefore, the dSPACE controller board is easy to use, consumer-friendly, well-known, convenient to implement the Matlab code for controlling factors in all situations and it is far simpler and smooth to put into effect. The remaining paper is arranged sequentially such that Section 2 describes the PEHS open loop for converter/controller design, Section 3 presents the proposed hybrid optimization approach PIVC for PEHS converter, Section 4 illustrates on detailed simulation model of a PEHS converter using the proposed hybrid optimization method, Section 5 explains the hardware implementation setup, Section 6 describes the results and lastly Section 7 provides the conclusions of the study.

## 2. PEHS Open Loop for Converter Controller Design

This study presents the PEHS converter's design process utilizing the dSPACE DS1104 controller board.

### 2.1. Controller

The capabilities of the step-up converter rely on the performance of controller. Usually, the PIVC has been used in several regulatory applications because of its simplicity and low-cost maintenance, robustness, and ruggedness. PEHS comprises two parts, namely the mechanical part and electronic part. The mechanical vibration source oscillates the cantilevered piezoelectric energy harvester (PEH) and it generates a force on the piezoelectric surface. PEH (i.e., vibration transducer) produces an AC signal at its output during the oscillation, which is at a resonant frequency. Since electronic equipment require a steady DC voltage; hence, the output from the energy harvester must be rectified and controlled according to the required application. The overall open loop system of PEHS processes schematic diagram is shown in Figure 2.

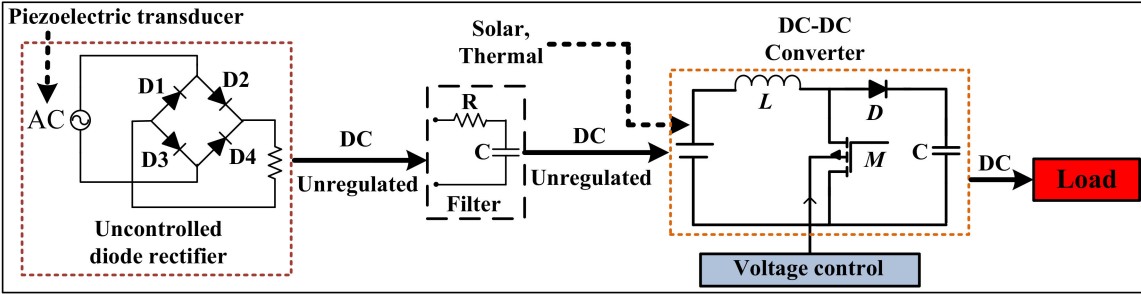

**Figure 2.** A conventional schematic diagram of PEHS [6].

### 2.2. Step-Up Converter Control Technique for PEHS

The goal is PEHS converter to regulate the DC output voltage at a favored small degree sign of numerous importance and frequency with oscillations and harmonic alteration. This regulation plays via a right controller imposing with regulator approach to keep the voltage at a set reference. The role of the duty cycle is essential to manage the converter parameter values. It is utilized to regulate the converter output voltage via tracking the reference voltage signal. The capacity of PEHS converter regulator system depends on the tuning reference value.

#### 2.2.1. Step-Up Converter Control

The single-phase open and close loop boost converter is a famous power electronic converter for PEHS. The boost converter is referred to as boost, buck, fly-back or buck-boost. The precept of this type of converter is the output voltage will increase or decrease, from the input source [32,33]. Numerous enhanced converter develops have evolved in the past few years. Recently, researchers have become increasingly interested in step-up converters due to the development of a wide variety of applications with strong supply voltages. A boost converter is an electronic converter with an output voltage designed to be upper than that of the input voltage. Input supply for the boost converter requires DC resources, such as photovoltaic, batteries, solar cells, rectifiers and DC turbines and these resources should oscillate because of adjustments inside the input voltage [34–37]. A capacitor is used as a filter to reduce the wave of the uncontrollable DC input voltage. A manipulate circuit is used to modify the DC output voltage at a preferred stage.

In an open loop action, the step-up converter shows low voltage control and inadequate energetic feedback, and such converter is utilized as a closed loop remark manipulate machine for the better output voltage. The closed-loop model converter has numerous benefits compared to the open-loop model converter. The principal merit of a closed-loop device is the capability to decrease a machine's impressibility to outside instabilities. Closed-loop structures are developed to robotically obtain and keep the favoured output circumstance with the aid of evaluating it with the real situation. It does this by producing an error signal, which has a variance of in the output and the reference input. On the other hand, a "closed-loop machine" is a complete computerized machine that manipulates movement reliant on the output in some manner. Therefore, with the aid of structuring a closed-loop, it reduces errors by automatically adapting the structure's input, enhance constancy of an unsteady structure as well as generate a dependable and repeatable conduct. A few authors have suggested exchanging the converter's on and off (duty cycle) to obtain the most suitable outcome from the regulator. Power stream is controlled over excessive frequency PWM for switching the MOSFETs in the circuitry of the step-up converter [38,39]. The merit of the single- stage step-up converter is the requirement of a signal MOSFET to be exchanged because its power losses are minor and is appropriate for the PEH packages.

### 2.2.2. PWM Switching Technique

Various techniques have been introduced to model DC-DC converters presented the PWM switch model with a circuit oriented approach. PWM of a signal includes the variation of its duty cycle to regulate the amount of power transfer to the load. It involves switching at constant frequency and adjusting the close period of the switch to regulate the average output voltage. The regulated signal, which is produced through comparing a signal level regulated voltage with a sawtooth waveform.

The switch regulates signal at PWM mode, which regulates the (close and open) of the switches, is produced through comparing a signal-level control voltage $V_{control}$ (i.e., error voltage $V_e$) with a constant waveform as appeared in Figure 3a,b. The regulated output voltages are usually achieved through the fault and the variation among the real voltage outcomes or its reference value. The frequency of the constant waveform with a steady peak, is a sawtooth type waveform creates the switching frequency. This frequency provides a stable PWM control signal between kilohertz to hundred kilohertz. When the error signal is bigger than the sawtooth waveform, the switch regulated signal becomes high, in this case the switch is closed, then, the switch open again.

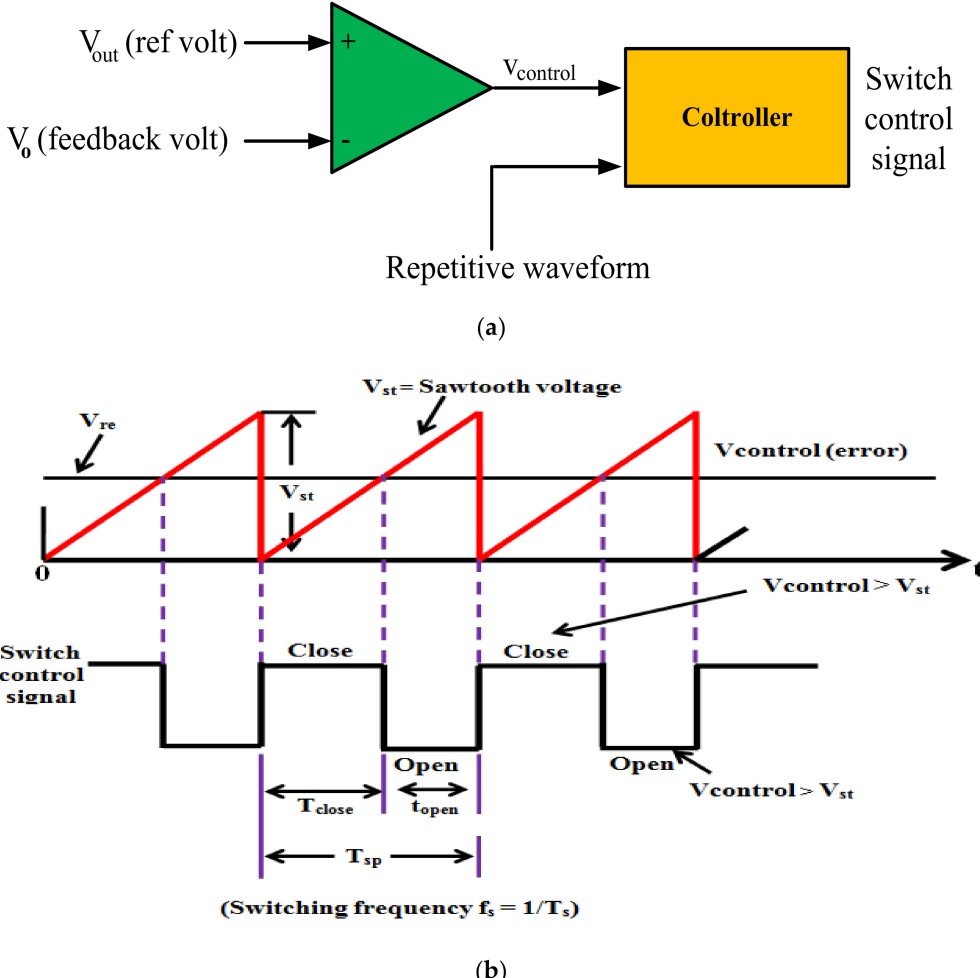

**Figure 3.** Pulse-width modulator (**a**) block diagram; (**b**) comparator signals.

### 2.3. Step-Up Converter Control Technique for PEHS

The capability of connecting the MATLAB/Simulink model to the real hardware has made the dSPACE DS1104 suitable for control platform. The dSPACE-based control system is performed through proposing the dSPACE Real-Time Interface (RTI) library blocks into the MATLAB/Simulink converter model. Through the MATLAB/Simulink real-time workshop (RTW) tool, a Simulink model with the dSPACE real-time blocks is converted to

C-codes automatically and starts the hardware operation. The DS1104SL_DSP_PWM block is utilized to produce the single-phase switching signals for the power devices [40].

The graphical user interface (GUI) is an application of the dSPACE, that the observation of operation and conduct of the converter in actual time is formed potential. Besides, the user is capable to modify the controller parameters and instantly monitor the effect of the system performance in a real time as well. In order to have a real-time converter operation system whereby the simulation is run in real time, a dSPACE DS1104 controller is required. Figure 4 demonstrates structure of the dSPACE DS1104 controller board. It displays the overall connection of the regulator board with the PC and converter equipment [40].

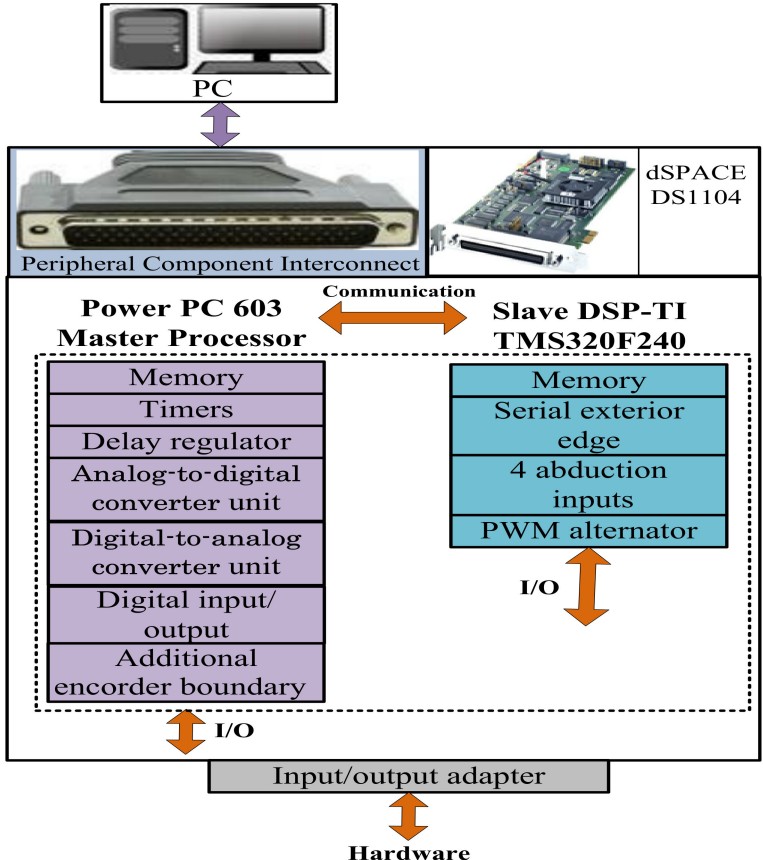

**Figure 4.** Structure of the dSPACE DS1104 controller board.

*2.4. Control Strategy*

The flow chart of dSPACE controlled converter defined in this work is presented in Figure 5, that shows a piezoelectric vibration transducer (PVT), full-wave AC-DC rectifier circuit, low pass filter, step-up converter, a dSPACE DS1104 board, and resistive load [41]. The response loop is used for associating voltages outcomes of the converter over the dSPACE regulator board. The operation of the MATLAB/Simulink power converter regulator method simulated in progressively is performed utilizing a dSPACE real time interface. Moreover, the dSPACE library function is essential in the regulator technique [42].

The analog-to-digital converter (ADC) channel of the controller dSPACE board is utilized for added signal coping. With a standalone step-up power converter, the access is to regulate the outcome of voltage provided to the resistive load. The obligation proportions of exchanging device are improved through the regulate wave that is the chosen essential frequency of the converter outcome. The triangular signal of 10 kHz generates the switching signals to handle the MOSFET. Each signal such as triangular signal, converter switching frequency is regulated through the switched MOSFET.

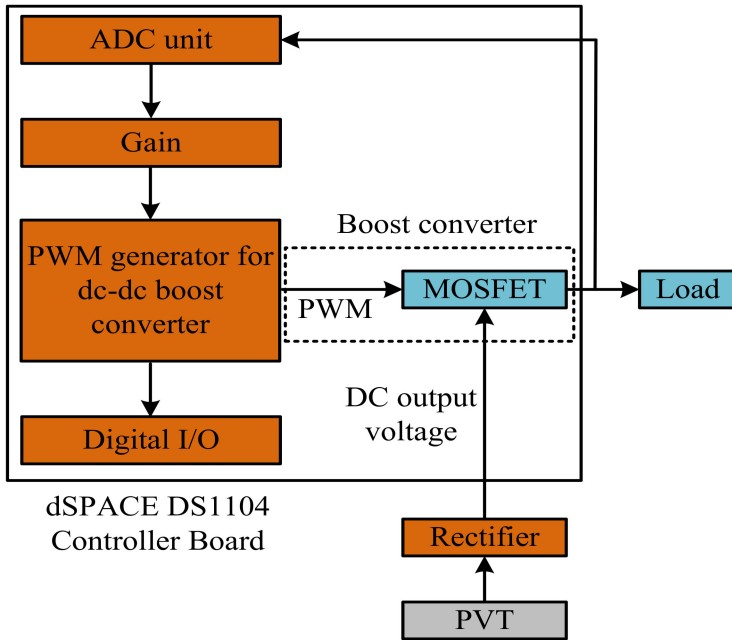

**Figure 5.** Flow chart of the dSPACE DS1104 controller unit.

## 3. Proposed Hybrid Optimization Approach PIVC for PEHS Converter

The previous unaided PEHS is unpredictable and may incur in unstable behavior when applied to a real application. The hybrid optimization technique is initiated to expand the overall PEHS execution. PVT is distinguished by the capacity to change unused vibration energy into charge. In any case, this transducer cannot work separately since it produces a small charge. By combining a PVT and PIVC, a hybrid optimization technique is developed to defeat this drawback of PEHS. The standard PIVC develop process needs mathematical modelling and a trial-and-error method. However, the complex segment of PIVC is to find the suitable parameter values $K_p$, and $K_i$. Previously, manual tuning was needed to acquire parameter estimations of controller and that was time consuming. Thus, the present work proposed a methodology to optimize $K_p$ and $K_i$ parameters values utilizing a hybrid optimization approach. Moreover, it is based on a simple model, with low maintenance cost, and does not rely on any mathematical model.

### 3.1. Optimum PIVC Design Based on LSA

The optimization algorithm consists of three necessary structures, counting input information, a fitness function, and optimum boundaries. Each component is performing for enhancement and organization to find best optimum values of PI controller parameter. The hybrid method is to get the suitable arrangement through reducing the fitness function by the information data and the selection of the drawback in each population of the repetitive process.

#### 3.1.1. Input Information

The first step in PIVC model, the values of $K_p$ and $K_i$ are presented to produce the solve from the hybrid optimization approach. Relaying on the number of parameter values of PIVC, the input vector $Y$ can be explained by:

$$Y_{i,j}\left[Z_{i,j}^1 Z_{i,j}^2 \ldots Z_{i,j}^n\right] \tag{3}$$

where $Y_{i,j}$ denotes the $j_{th}$ solving in the generation through the $i_{th}$ iteration, and $n$ is the whole quantity of parameters. In this paper, two-issue measurements and a population of 50 have been estimated to get the appropriate parameter values of the PIVC.

### 3.1.2. Fitness Function

A fitness function is required for the hybrid optimization technique to obtain a reduced error. Hence, the fitness functions search the suitable value for the PIVC outcomes to enhance the structure stability. The mean absolute error (MAE) is used as a fitness function to search best parameter values of controller for suitable outputs. The MAE operation might be counted as represented in Equation (4):

$$\text{Fitness function (MAE)} = \frac{1}{S}\sum_{i=1}^{S}|error| \tag{4}$$

where, $S$ is the value for sample, error is PIVC for step-up converter. In the hybrid optimization procedure, Equation (4) requires to be minimized.

### 3.1.3. Optimization Constraints

The hybrid optimization technique can be executed to define the best values of the $K_p$ and $K_i$ parameters. The boundaries of these parameters must not overlap. Besides, the element $Z_{i,j}^k$ must be between $Z_{i,j}^{k-1}$ and $Z_{i,j}^{k+1}$. If the element $Z_{i,j}^k$ is higher than $Z_{i,j}^{k+1}$ or smaller than $Z_{i,j}^{k-1}$, this element might be reconstruct inside its boundaries. Therefore, this constraint should be agreed to ensure that every $K_p$ and $K_i$ parameters are inside the applicable boundaries:

$$Z_{i,j}^{k-1} < Z_{i,j}^k < Z_{i,j}^{k+1} \tag{5}$$

### 3.2. LSA Theory

LSA is a relatively new optimization technique proposed by [30]. This approach consists of three stages; shell and pace frontrunner spread, shell feature, and shell exhibiting and measure. The shell period is related to the particle period in particle swarm optimization (PSO) technique and agent term in the backtracking search algorithm (BSA) technique. The shell velocity is determined via:

$$v_p = \left[1 - \left(\frac{1}{\sqrt{1-(v_0/l)^2}} - \frac{pF_i}{mc^2}\right)^{-2}\right]^{\frac{-1}{2}} \tag{6}$$

where $v_p$ is current velocity of the shell; $v_o$ vital velocity of the shell; $F_i$ is the stable acceptance proportion, $l$ is the light velocity; $m$ is the mass of the shell; and $p$ is the length of the path migrate.

The LSA method generates splitting through two techniques, the first technique is a process to generate regular terminals due to the centers impact of the shells which are recognized through utilizing the converse number as shown below:

$$\bar{p}_i = c + d - p_i \tag{7}$$

where $\bar{p}_i$ is opposite projectiles in one-dimension, $c$ and $d$ are edge boundary, and $p_i$ is an original projectile in one-dimension.

There are three categories of projectiles to present the alteration shells that produce the first-pace frontrunner generation. An alteration shell could be an arbitrary way by the evolution forms an evicted projectile from the noise unit. Hence, an alteration projectile might show an arbitrary number through the development of the arbitrary channel in the area.

### 3.3. LSA to Achieve the Ideal PEHS Converter

The operation begins by readjusting the LSA parameters, namely the values of iterations ($I$), population proportion ($P$), issue element ($M$), $E_p$ shell energy, $E_{sl}$ pace frontrunner energy and process time. The proportional integral (PI) controller is taken into consideration as

one of the control strategies proposed in this PEHS. The PI set of rules tuning entails the calculation of two essential different parameters, which consist of the proportional and integral modes. Proportional control minimizes the error, and integral control eliminates the balance. Moreover, the PI regulator is utilized to control voltage, current, and numerous others. Figure 6 shows the operation design of a PI regulator which obtains a minimum signal for a regulator pointer alongside large overshoot, extreme steady-state mistake, and force. Thus, this regulator generates a voltage outcome signal which consists of total of blunders with the vital subsidiary, and corresponding of that mistake, as appeared in the Equation (8):

$$u(t) = K_p e(t) + K_i \int\limits_0^t e(t)dt \tag{8}$$

where $e$ is the error $e = (X_{reference} - X_{measure})$, $u$ is the regulator outcome indicator, $K_p$ is the proportional gain, and $K_i$ is the integral gain. The presentation of the PI regulator essentially relies upon the selected appropriate PI boundaries. Every boundary assumes a significant part in converter regulating the PEHS as shown in Table 1.

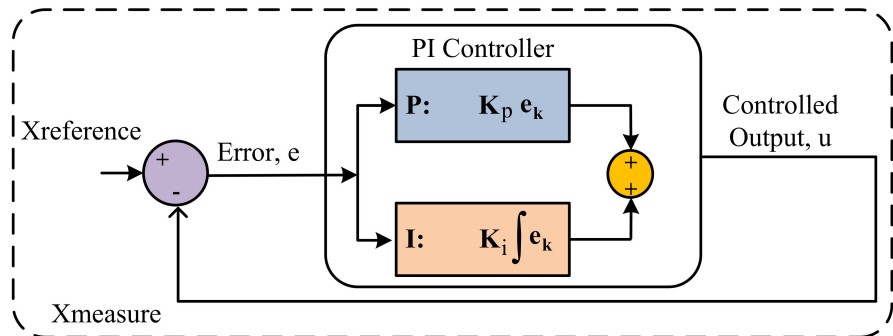

**Figure 6.** Structure of proportional integral (PI) control.

**Table 1.** Attributes of proportional integral (PI) regulator boundaries.

| Type | Rise Time | Overshoot | Settling Time | Steady State Error |
|------|-----------|-----------|---------------|--------------------|
| $K_p$ | Lessening | Rise | Little change | Lessening |
| $K_i$ | Lessening | Rise | Rise | Reduce |

The primary populations of the $K_p$ and $K_i$ are formed and scheduled equivalent to Equation (5) generated. The opposite section contains an estimate of the fitness function via Equation (4). Later the primary population is predicted, the rule and location are modified with Equations (6) and (7), correspondingly. Subsequently enhancing every the estimations of $Z_{i,j}$ in the population, the produce the fitness function, and the activity operate to the following iteration. This enhancing and fitness function re-evaluate generate is a temporary process until the upper iteration amount is extended:

$$q^S_{i\_new} = q^S_i \pm \exp rand(\mu_i) \tag{9}$$

$$q^L_{new} = q^L + normrand(\mu_L, \sigma_L) \tag{10}$$

where, $q^S_{i\_new}$ is the current location shell, $q^S_i$ is the primitive location shell, and $q^L_{new}$ is the current governing shell. Later improving each amount of estimations of $Z_{i,j}$ in the populace, the create fitness function, and progress to the subsequent iteration. This enhancing and fitness function re-evaluation system is temporary up to the peak iteration ($I$), population size ($P$) and problem dimension ($M$), $i$=1 process time calculates is completed as described by the pseudo code for the LSA technique shown in Algorithm A1 (Appendix A). From this Algorithm A1 LSA starts to determine the generation of the primary pace frontrunner (alteration shell $z$) using Equation (5) and runs the simulation with PEHS for all $Z_{ij}$. We compute the fitness

function MAE, utilizing Equation (4). After updating the frontrunner points force, $E_{sl}$, top and lowest pace frontrunners, we update the kinetic energy, $E_p$, direction and location according Equations (9) and (10), running the system with each PEHS for each $Z_{ij}$ and computing the fitness function, utilizing Equation (4). Finally we generate two in proportion process at the fock place and return a lightning strike point (best pace frontrunner).

## 4. Simulation Model of a PEHS Converter using the Proposed Hybrid Optimization Technique

Figure 7 shows the developed MATLAB/Simulink PEHS converter with a PI voltage controller simulation model. The AC input voltage, $V_{in}$, of 300 mV generated from the PVT for the PEHS converter is supplied by the voltage source. The operation of the PEHS is as follows: the PVT generates an amount of energy from vibrations. The action of one side of the PVT is open and the other side is fixed. Once the force is applied on the fixed part of the PVT then the open side of the PVT vibrates and produces a small amount of unstable AC electrical charge with harmonics. Then a full-wave diode bridge rectifier is required to convert this into a DC charge. Still the amount of the rectified DC voltage is very small and unstable, therefore a boost converter with a controller is needed to expand the voltage level. The outcome of the boost converter has some drawbacks in terms of fluctuations, instability and difficulty to regulate for low power applications, so a robust converter controller is required to overcome this problem.

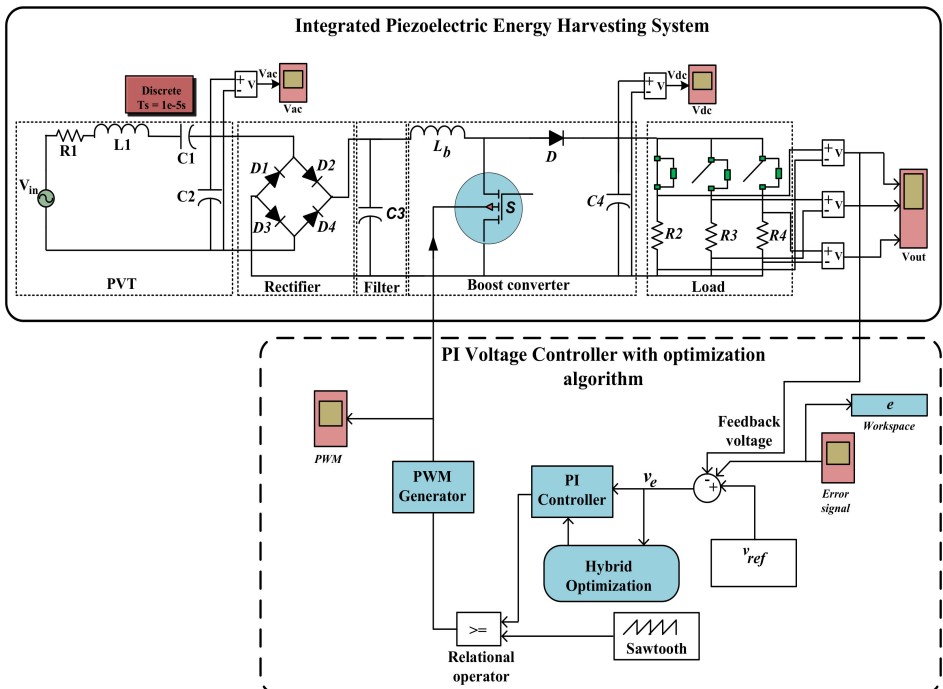

**Figure 7.** Schematic development of the proposed lightning search algorithm-proportional-integral voltage controller (LSA-PIVC) for the piezoelectric energy harvesting system (PEHS).

The voltage outcome is larger than the input voltage as represented in Equation (11). When $U$ duty cycle is lower than one, the outcome $V_{out}$ is upper than the input $V_{in}$. Accordingly, a step-up converter voltage can be increased, and the improved voltage amount is characterized by $U$. Theoretically, the duty cycle of MOSFET should be set in such a way that the DC output voltage of this converter is sufficiently high, around 7 V DC, for the PEHS converter to be able to produce an AC $V_{rms}$ input voltage of 300 mV. The common inductance of the step-up converter, $L_b$ is calculated by Equation (12). Here $L_b$

is the inductance of the converter and $f_s$ are the switching frequency. The entire power loss of the converter is determined by Equation (13):

$$U = 1 - \frac{V_{in}}{V_{out}} \tag{11}$$

$$L_b = \frac{U(1-U)^2}{2f_s} \tag{12}$$

$$P_{LS\_total} = P_{ML\_DS} + P_{DL\_D} + P_{L\_iL} \tag{13}$$

where $P_{LS\_total}$ is the overall power loss of converter, $P_{ML\_DS}$ is the MOSFET transmission loss, $P_{DL\_D}$ is the total diode conduction loss and $P_{L\_iL}$ is the inductor loss of the converter [43]:

$$P_{ML\_DS} = \frac{2r_{DS}}{3} \sqrt{\frac{2(U-1)^3}{f_s L_b U}} P_{in} \tag{14}$$

where, $r_{DS}$ is the resistance of the MOSFET:

$$P_{DL\_D} = P_{RF} + P_{VF} \tag{15}$$

here, $P_{RF}$ is the power loss in the diode forward resistance, $R_F$ is the diode forward resistance, $V_F$ is the diode threshold voltage and $I_P$ input current of the PZT and $r_L$ is the inductor resistor:

$$P_{RF} = \frac{2R_F}{3} \sqrt{\frac{2(U-1)}{f_s L_b U}} P_{in} \tag{16}$$

$$P_{VF} = V_F I_p \tag{17}$$

$$P_{L\_iL} = \frac{2r_L}{3} \sqrt{\frac{2U(U-1)}{f_s L_b}} P_{in} \tag{18}$$

The power efficiency of the converter is determined by Equation (19):

$$\eta = \frac{P_{in}}{P_{in} + P_{LS\_total}} \times 100 \tag{19}$$

The proficiency of the low power boost converter is primarily measured as 85% to achieve the converter boundaries with the goal that the whole converter power loss might be counted to get the real converter effectiveness. The overall process is conducted via the control structure where PWM strategy increase ideal $K_p$ and $K_i$ values utilizing hybrid optimization approach for PIVC and generate PWM signals to drive the MOSFET.

## 5. Hardware Implementation Setup

The experimental construction is the illustration of an electronic structure, to control the process operation of the layout for a prototype structure. The experimental operation is one of the major components to prove the system capability.

### *Experimental Setup*

The real-time simulation operation of the boost converter control method is completed through utilization of a DS1104 controller board from dSPACE. Therefore, the input-output library blocks of the dSPACE are necessary to operate the control method. Figure 8 presents the real-time converter control method hardware process via dSPACE controller board.

The dSPACE block consists of DS1104MUX_ADC, and DS1104SL_DSP_PWM. By using the DS1104MUX_ADC block, the converter output voltage, $v_a$ is multiplexed and fed into the controller. The DS1104ADC is used to channel the external parameters, e.g., voltage, currents, into the control system. Finally, the DS1104SL_DSP_PWM is used for the generation of PWM switching signal for the MOSFET.

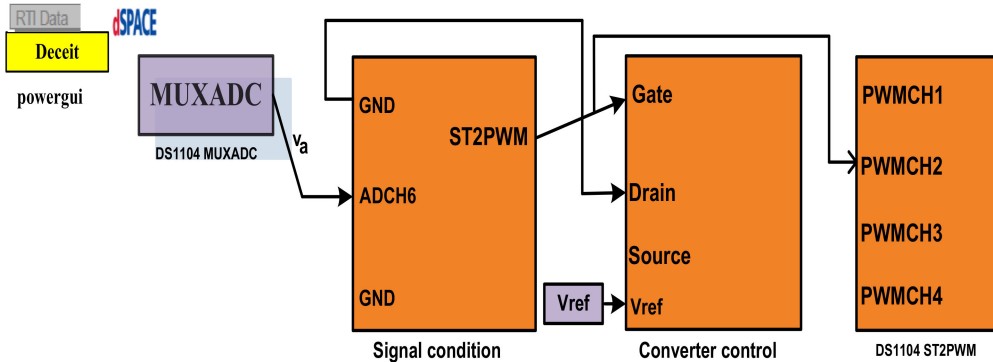

**Figure 8.** Real-time converter control hardware process via dSPACE.

The PVT spree is a pre-mounted and prewired twofold speedy mount twisting generator. The utilization of a vibration generator with a force vibration shaker is offered to regulate the mechanical energy creation by shifting the frequency. The block diagram and test arrangement for information extraction from the PVT is appeared in Figure 9. To lead the PVT investigation, the regular frequency has been changed between 10 Hz to 60 Hz through the amplifier unit.

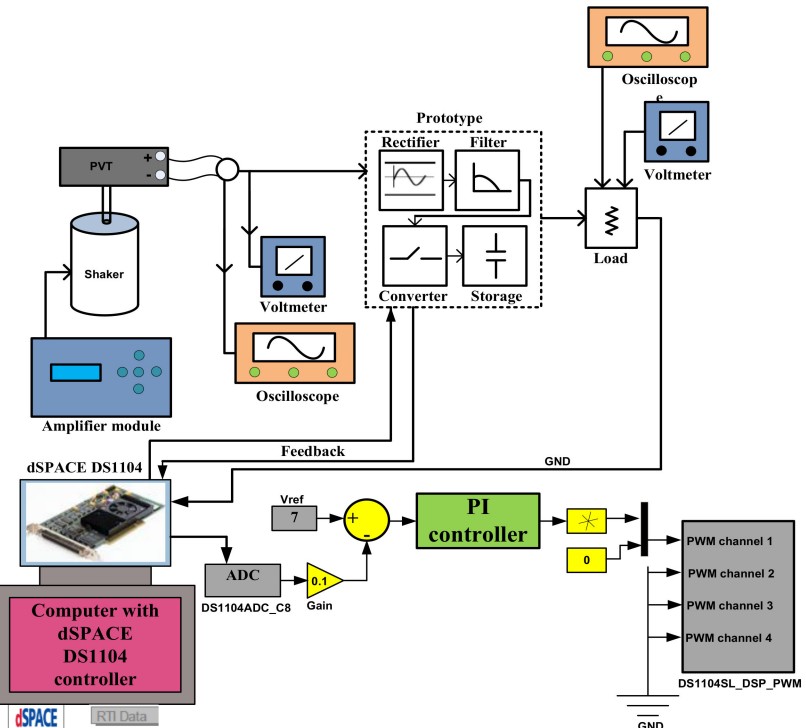

**Figure 9.** Block diagram of experimental test bench layout.

As part of an improved PEHS prototype, the overall action estimation is important to confirm the accuracy of the simulation. To confirm the simulated design, an incorporated PEHS prototype was built, tested, and assessed in the laboratory, as presented in Figure 10. To prove the simulation results, a PEHS experimental operation process was implemented in a printed circuit board layout in the laboratory.

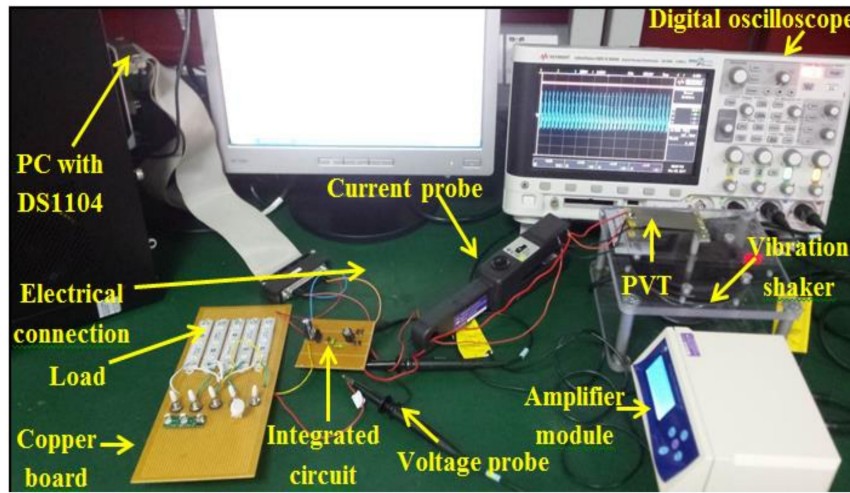

**Figure 10.** PEHS converter control process hardware setup with dSPACE.

## 6. Results

This part presents the optimum outcomes of hybrid optimization method and PI voltage controller approaches of the PEHS converter.

### 6.1. MOSFET PWM Switching Signals

A control signal is the main key for switching devices to control the converter. The PWM signal plays an important role to control the converter switches such as MOSFET ON and OFF, so the converter control efficiency depends on the smooth operation of the switching mode.

A snapshot of a few cycles of the MOSFET gate switching signal obtained from the oscilloscope is illustrated in Figure 11, which shows the 10 kHz switching frequency, where one cycle of the PWM signal is measured at 10 µs to show the hardware operation process of the PEHS converter. Finally, after analysis, the pattern of the signal shows that the duty cycle of the PWM is rising. It is noted that in the simulation, the regulator signal is in digital form, which is in the form of '0' and '1' level. The first one makes the switching device to turn ON, while the later one turns the switch off.

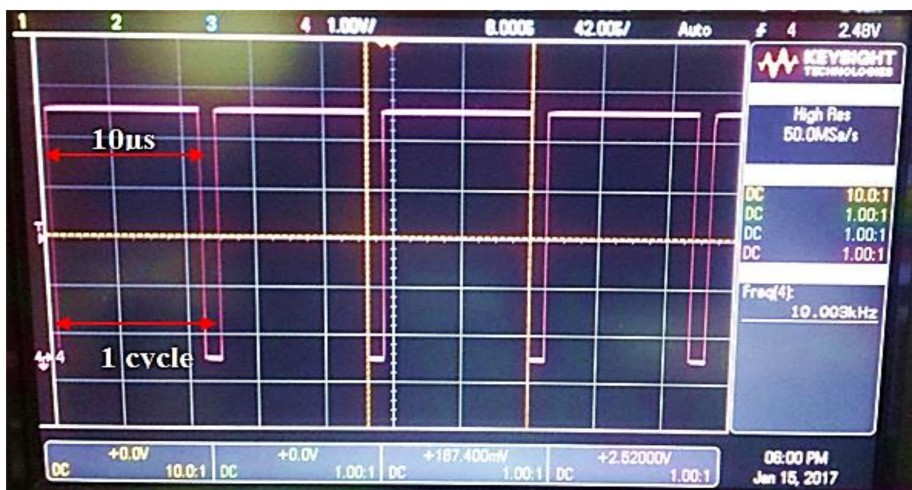

**Figure 11.** Oscilloscope snapshot of the pulse width modulation (PWM) switching signal for a MOSFET gate.

### 6.2. Controller Parameter Waveforms in dSPACE

Generally, the real-time implementation results are little changed compared to the simulation results described below. In the simulation the hardware results might be

modified through the uncontrollable behaviors of parameters, e.g., element resistance difference and circuit-dependent inductance and capacitance which involve the actions of the converter, particularly the filter. However, the denoted results are measured as good according to the match between the simulation and experimental data.

To identify the converter system operation in real-time, a GUI has been developed using the software from dSPACE called ControlDesk. The nature of the converter and control system parameters are output voltage waveforms, correspondingly, they can be examined in real-time. Having the merit of connecting to the control technique in Matlab/Simulink environment, the effect of any parameter value changes on the technique can be seen simultaneously in the layout of the GUI. In the dSPACE DS1104 system, the input parameters level must be less than 10 V. Figure 12 presents an example of output feedback voltage per unit, obtained from the output terminal of the measurement PEHS converter circuit. This voltage is considered to be the feedback signal to be processed in the dSPACE-based control system. It presents a feedback voltage of approximately 7.1 V DC, which is less than 10 V.

Figure 13 presents a computer screen snapshot of some of the examples of the parameter which are organized in a simple layout in the GUI. These are the real-time parameters which denote the performance and nature of the converter prototype and its controller. Some of these parameters are the measured output voltage, reference voltage, feedback voltage and their corresponding errors.

*6.3. Simulation and Analysis for Fitness Function MAE with Various Loads*

The enhanced hybrid optimization method-based PIVC for PEHS converter output results is presented in this section. The simulation results were compared with the experimental consequences to confirm the capability of the proposed PEHS converter. The PIVC parameters were correctly measured through the LSA technique to find the most suitable $(K_p, K_i)$ boundary conduct for the PIVC. This has been carried with the BSA and PSO approaches to determine the optimal PI boundary, in comparison to LSA. The assessment aims to evaluate and verify the toughness, exactness, strength, and effectiveness of the LSA optimization. Figure 14 presents the optimization outcomes of the LSA, BSA and PSO using the MAE fitness function for 500 iterations. The LSA method achieves a smaller fitness function for the MAE values. The results presented in Table 2 are the best, worst, median, average, and standard deviation of the fitness function of MAE and LSA offers the best performance for 2 MΩ, 100 kΩ, 470 Ω and 50 Ω loads. These are the most important linear loads that affect the system. Therefore, the efficiency of the controller appears clearly when these loads are used in terms of the time in the simulation and hardware matches closely almost the same time, and the controller in the hardware has enough time to regulate the waveforms. The best outcome for every load is mentioned in boldface. To control the consistency and overall performance of the LSA, the information detected in the box plot is illustrated in Figure 15. Table 3 presents the estimations of the outcome error by the three different techniques. Table 3 clearly shows that the LSA method produces the lowest errors in the minimum boundary estimations of the PIVC and regulates the PWM signal so that the responsibility of the controller's PWM outcome shows toughness, sturdiness, and efficiency. To confirm that the comparisons are clear, an identical population size and maximum amounts of iterations were implemented in all methods, as presented in Table 4.

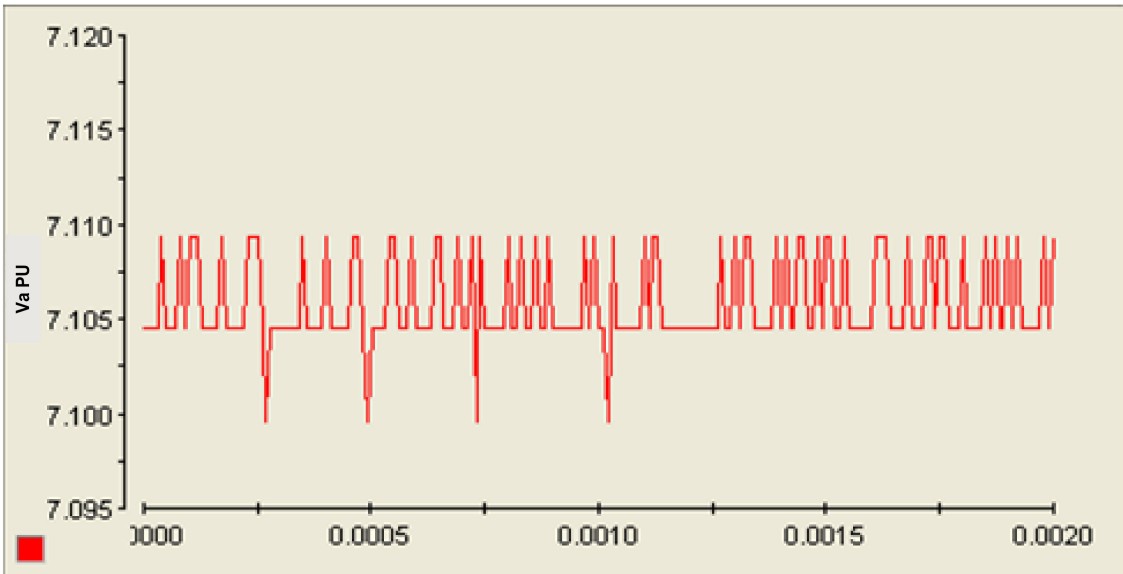

**Figure 12.** Output measured voltage per unit from dSPACE graphical user interface (GUI).

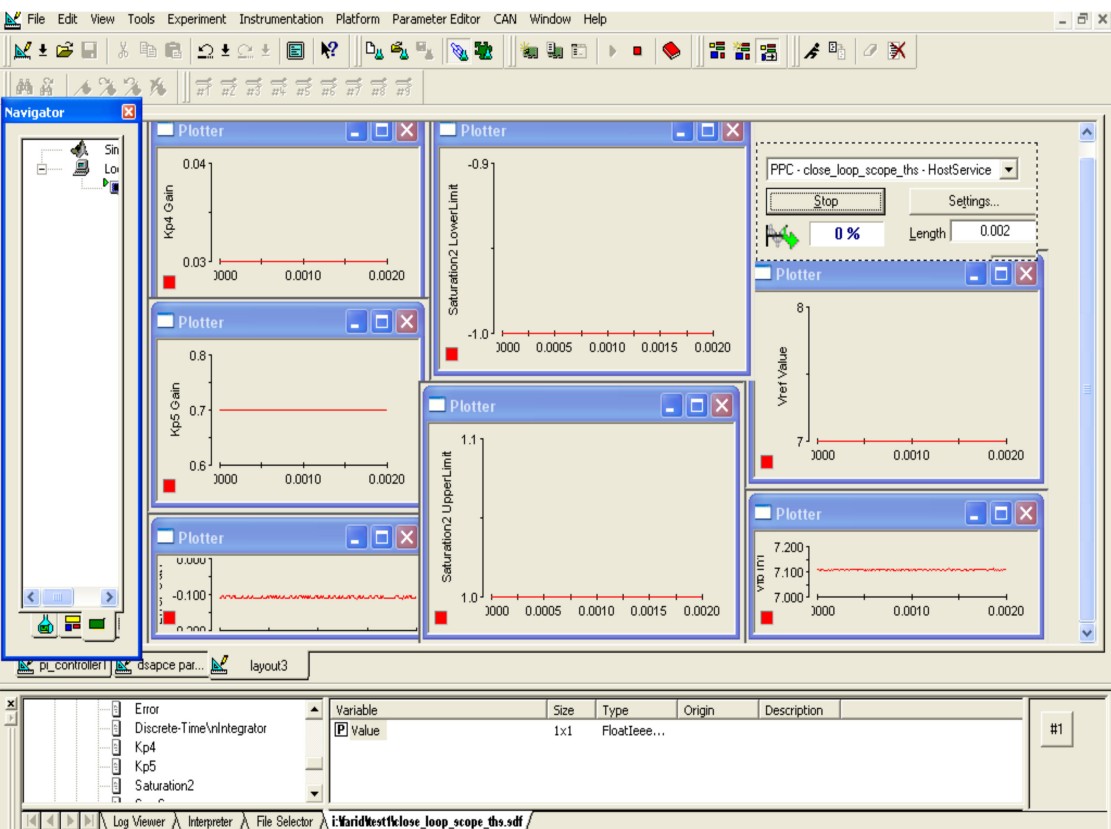

**Figure 13.** dSPACE GUI parameter layout.

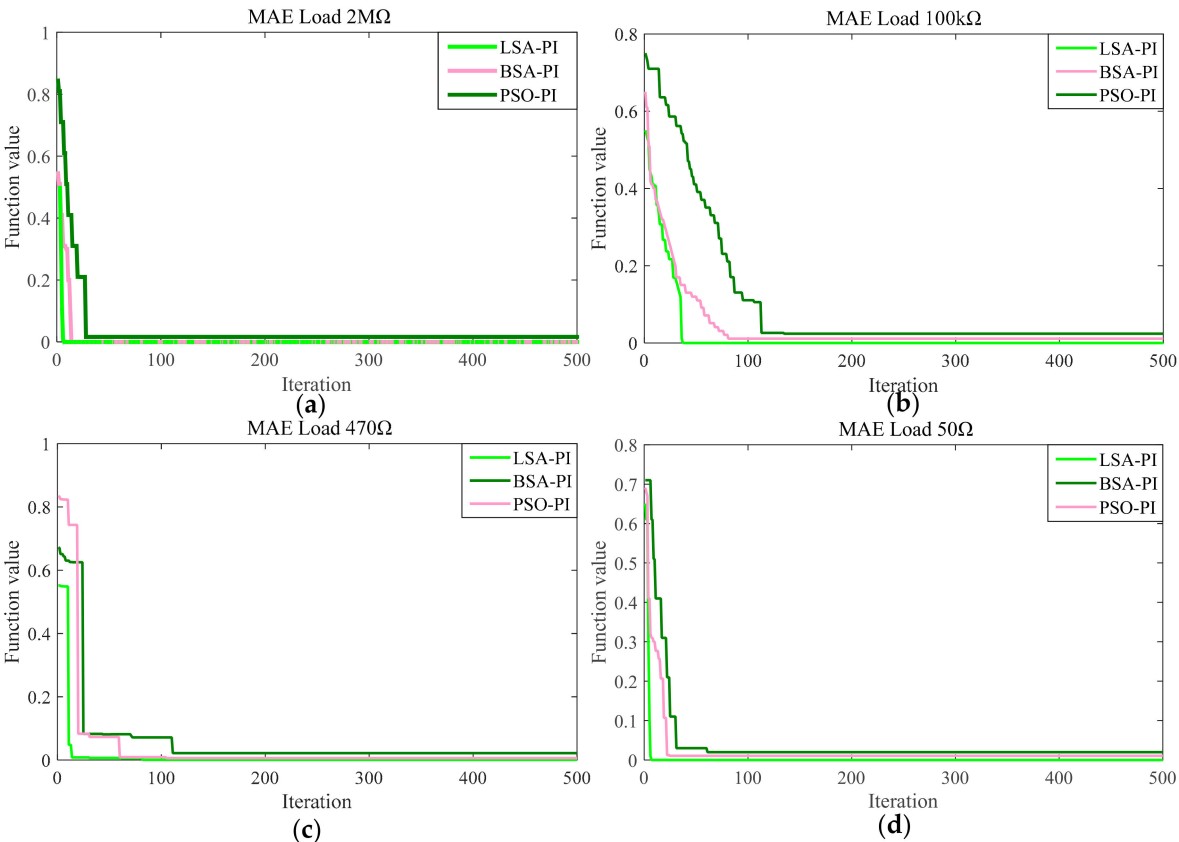

**Figure 14.** Convergence characteristics of optimization methods based on LSA-proportional integral (PI), backtracking search algorithm (BSA)-PI and particle swarm optimization (PSO)-PI for fitness function mean absolute error (MAE) with load (**a**) 2 MΩ; (**b**) 100 kΩ; (**c**) 470 Ω; (**d**) 50 Ω.

**Table 2.** Global optimum outcomes for every fitness functions in mean absolute error (MAE).

| Load | Measurements | LSA | BSA | PSO |
|------|--------------|-----|-----|-----|
| | Best | **0.000000000077343** | 0.0001633201 | 0.000657771 |
| | Worst | **0.00051020** | 0.030612321 | 0.031732138 |
| 2 MΩ | Average | **0.000008150** | 0.006876100 | 0.030834701 |
| | Median | **0.0000023007** | 0.003061005 | 0.020081260 |
| | Standard Deviation | **0.000071341** | 0.015217271 | 0.009188011 |
| | Best | **0.0000007399** | 0.001786800 | 0.001026139 |
| | Worst | **0.007010000** | 0.041251135 | 0.021731300 |
| 100 kΩ | Average | **0.000251000** | 0.007970010 | 0.00191980 |
| | Median | **0.0000105161** | 0.001926101 | 0.012100108 |
| | Standard Deviation | **0.000912100** | 0.026190084 | 0.019500431 |
| | Best | **0.00014500** | 0.019419001 | 0.003190087 |
| | Worst | **0.10107936** | **0.211004200** | 0.12273209 |
| 470 Ω | Average | **0.008223118** | 0.023110209 | 0.00910008 |
| | Median | **0.00062002** | 0.019612906 | 0.010021490 |
| | Standard Deviation | **0.0091101006** | 0.031000031 | 0.041000120 |
| | Best | **0.0000009110** | 0.020005160 | 0.023009811 |
| | Worst | **0.030023418** | 0.080200145 | 0.893018799 |
| 50 Ω | Average | **0.002811000** | 0.030175100 | 0.20019932 |
| | Median | **0.0000026513** | **0.00081002** | 0.003001010 |
| | Standard Deviation | **0.006109510** | 0.030015951 | 0.400110010 |

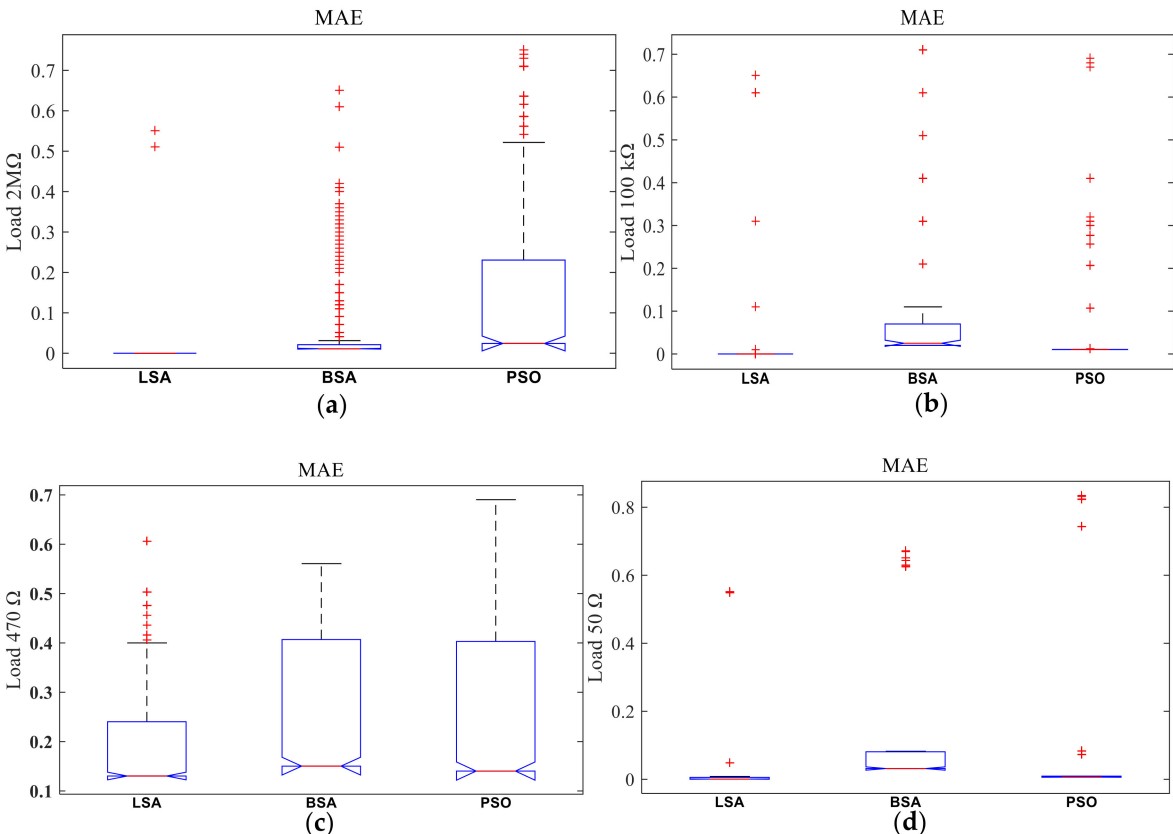

**Figure 15.** Variety in global optimum outcomes with MAE for load (**a**) 2 MΩ; (**b**) 100 kΩ; (**c**) 470 Ω; (**d**) 50 Ω.

**Table 3.** Estimations of outcome errors by different techniques.

| MAE | LSA | BSA | PSO |
|---|---|---|---|
| | 0.000000000077343 | 0.0001633201 | 0.000657771 |

**Table 4.** Boundary settings utilized in each optimization approach.

| Parameter | LSA | BSA | PSO |
|---|---|---|---|
| Population Size | 50 | 50 | 50 |
| Max. Iteration | 500 | 500 | 500 |
| c1 and c2 | - | - | 1.5 |
| F | - | 3 | - |
| Channel time | 10 | - | - |

### 6.4. Simulation and Hardware Outcomes in Terms of the Settling Time and Rising Time

This study analyzes the simulation and experimental results in terms of settling and rising time of PIVC using 300 mV as an input voltage to test the outcomes of the proposed PEHS converter using a hybrid optimization approach. The fitness function, as presented in Equation (4), is optimized through the LSA-PI based voltage controller when overcoming the limitations in Equations (5)–(7). To validate this, the performance of LSA-PI voltage controller is compared with BSA-PI and PSO-PI in terms of rise and settling time to confirm the stability of the proposed controller. The feedback of the PEHS converter simulation results based on hybrid optimization approach utilizing the LSA-PI, BSA-PI and PSO-PI voltage controller is presented in Figure 16. From Figure 16, it is observed that LSA-PI voltage controller for PEHS converter achieves the best optimal results utilizing a hybrid optimization approach in terms of rise and setting time compared to the others techniques (BSA-PI and PSO-PI) as the settling time is comparatively large for BSA-PI and PSO-PI.

The prototype hardware output results agree with the simulation outcomes as represented in Figure 17. Table 5 presents the comparison of the optimized results by utilizing these methods. The LSA-PI has effectively achieved the most suitable output compared with the other BSA-PI and PSO-PI controllers based on the rising time and settling time.

**Table 5.** Comparison outputs acquire utilizing hybrid optimization and PI controller.

| Techniques | Rise Time (s) | Settling Time (s) | Peak over Shoot | Consistent State Error |
|:---:|:---:|:---:|:---:|:---:|
| LSA-PI | 0.03579 | 0.0457 | 0% | 0.1 |
| BSA-PI | 0.05176 | 0.1213 | 0% | 0.1 |
| PSO-PI | 0.08235 | 0.1675 | 0% | 0.1 |

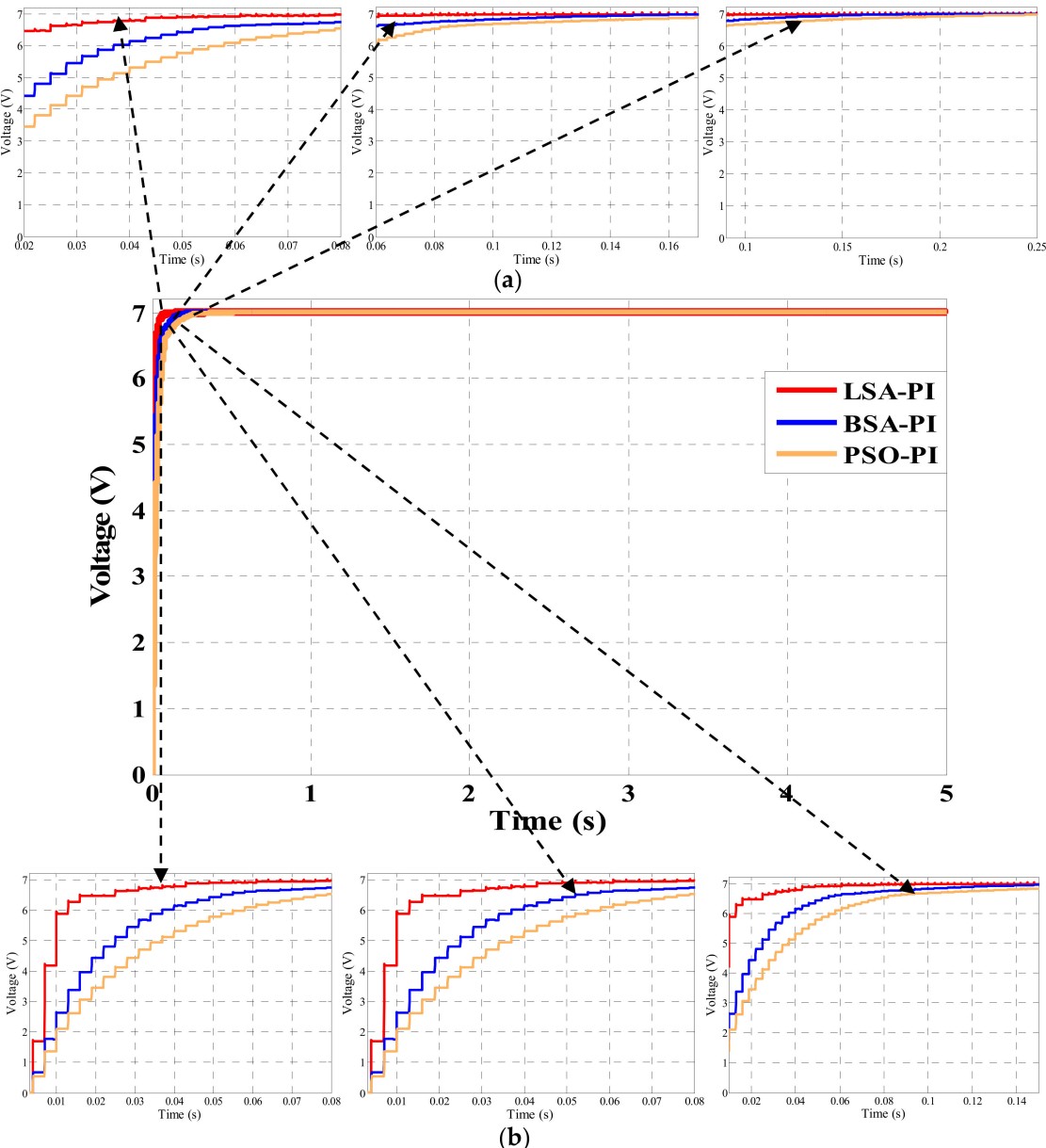

**Figure 16.** Simulation results hybrid optimization and PI voltage controller approach with rise time (**a**) and settling time; (**b**) for a 300 mV input.

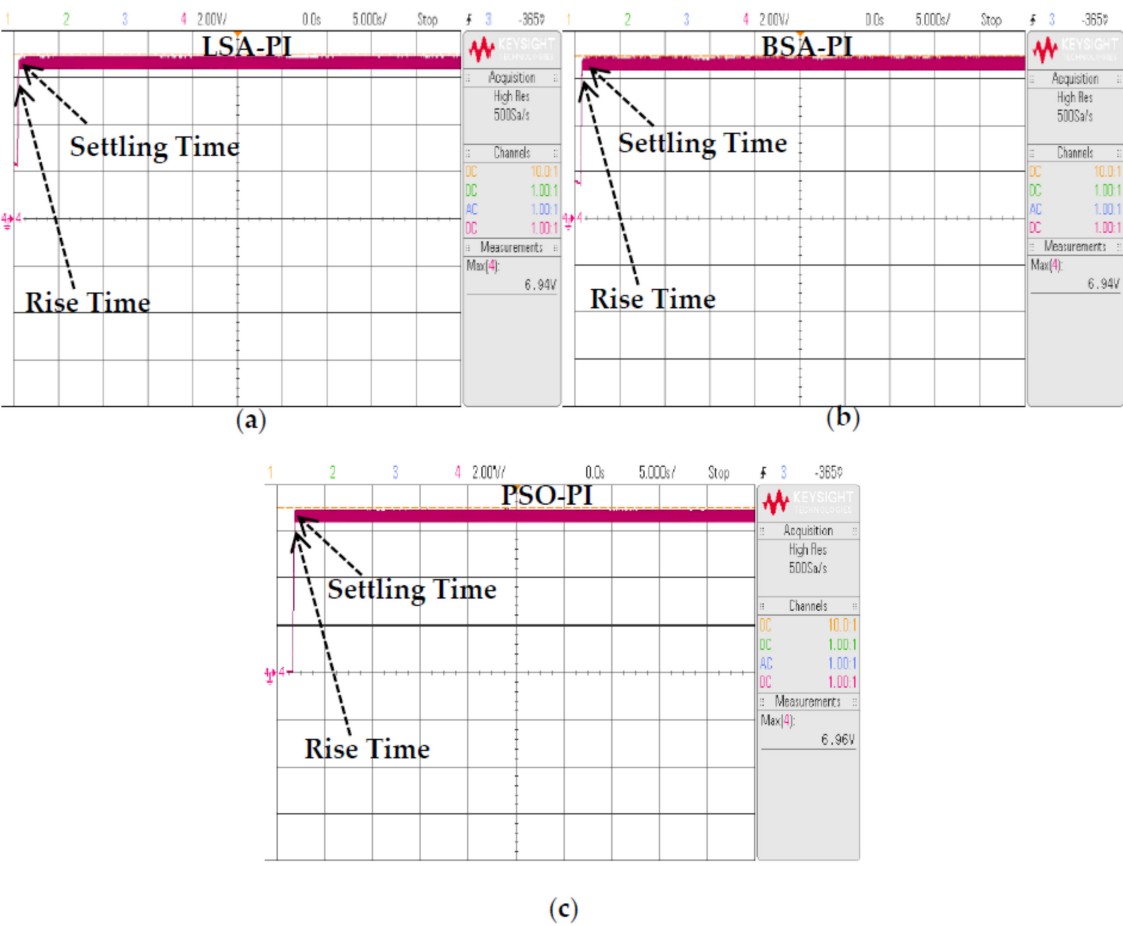

**Figure 17.** Experimental output LSA-PI (**a**), BSA-PI; (**b**) PSO-PI; (**c**) controllers considering rising and settling time.

*6.5. Loss Analysis and Simulation, Hardware Results of PEHS*

This study investigates the simulation and hardware output of a combined PEHS input and output voltage using optimization methods. The simulation results utilizing the LSA-PI, BSA-PI and PSO-PI techniques are shown in Figure 18a–c, respectively, that exhibit the input of a sinusoidal AC $V_{rms}$ = 300 mV, formed from the PVT, and an optimal DC voltage outcome of 7 V, using the LSA-PI, BSA-PI and PS0-PI methods. Figure 18a shows that at the initial stage the curve is bending but after 0.0457 s, the curve becomes steady. Moreover, Figure 18b,c show that the output voltage features of the BSA-PI and PSO-PI approaches are nearly related without any important modification for a change in input and load but the curve is bent; primarily, a delay is necessary to gain the voltage, but by using BSA-PI, the curve is constant after 0.1213 s, whereas by using PSO-PI, the curve was stable after 0.1675 s. The peak overshoot value was similar for the three techniques. Therefore, it could be presumed that the regulator's boundary values, achieved from LSA-PI, are suitable for the response PI voltage controller model. The prototype hardware result measurements were in good agreement with the simulated outputs, as shown in Figure 19a–c. The measured input current of the PVT was about 1.4 mA, as shown in Figure 20. The comparison between simulation and hardware output of the PEHS converter is shown in Table 6. The estimated and measured parameter values of the prototype are presented in Table 7.

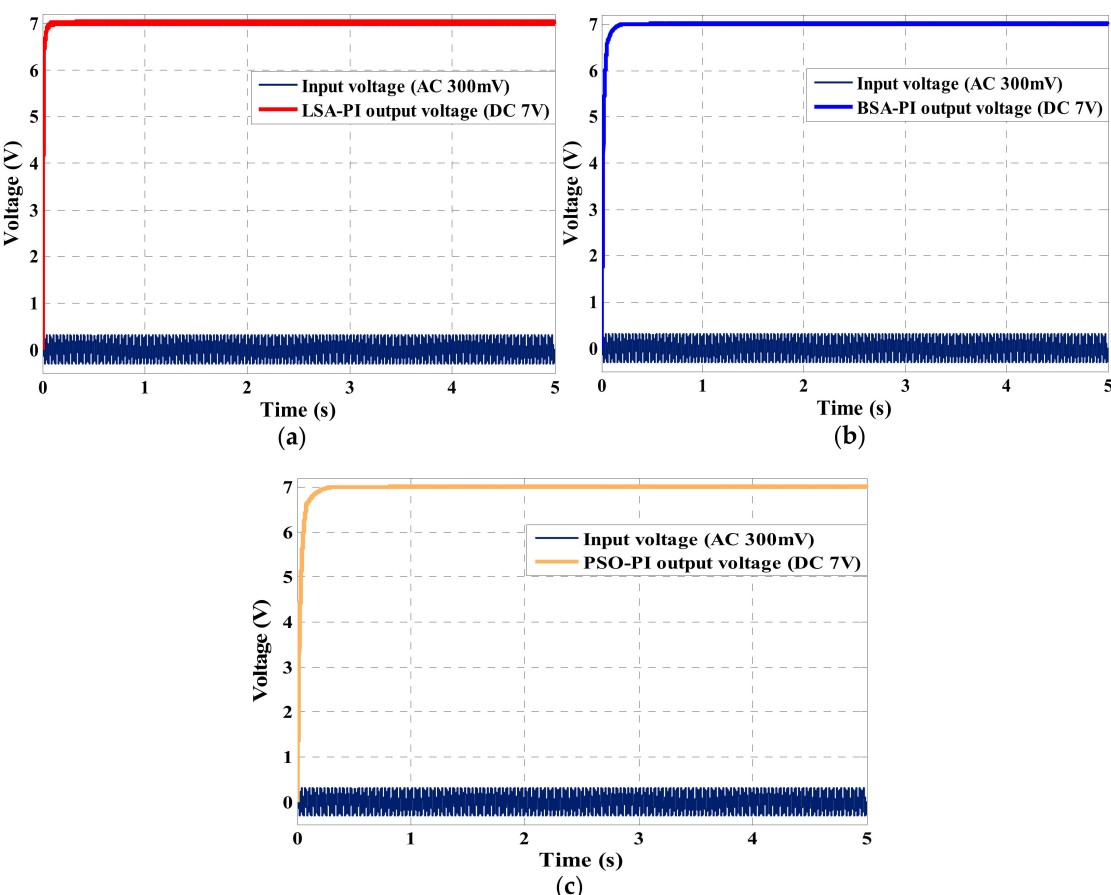

**Figure 18.** Simulation outputs of a PEHS using a LSA-PI (**a**), BSA-PI; (**b**) PSO-PI; (**c**) controller with input 300 mV.

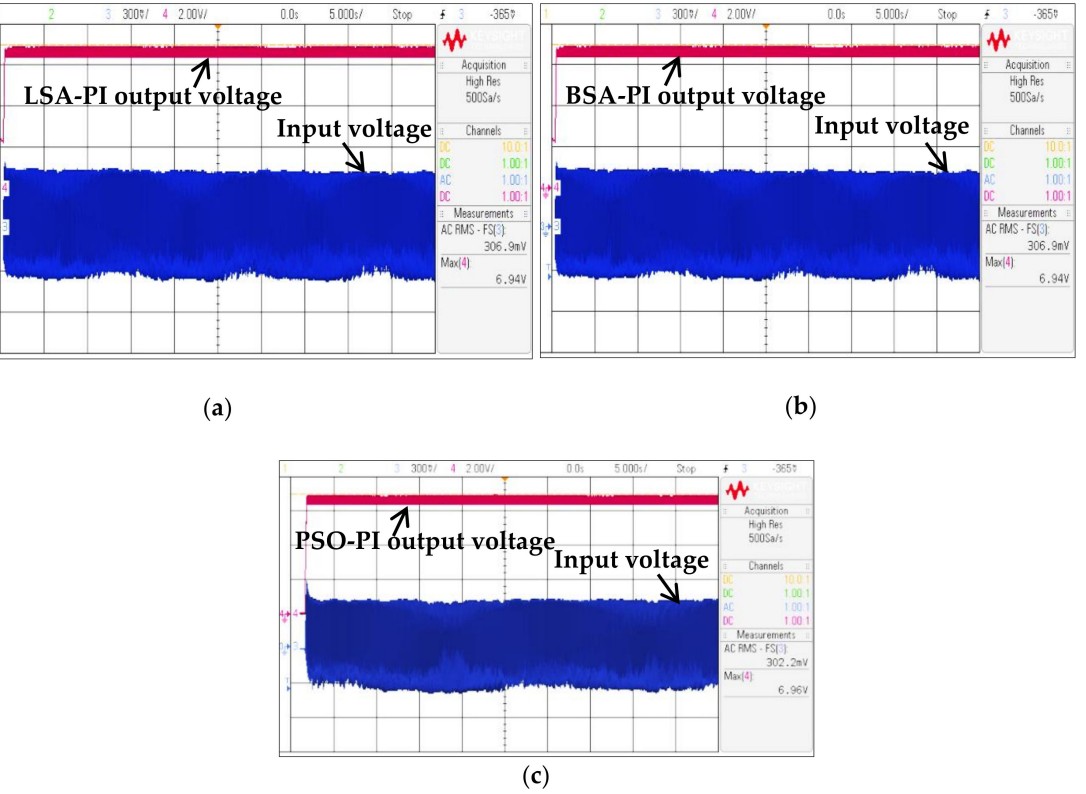

**Figure 19.** Experimental outputs of the PEHS converter using a LSA-PI (**a**) BSA-PI; (**b**) PSO-PI; (**c**) controller with input 300 mV.

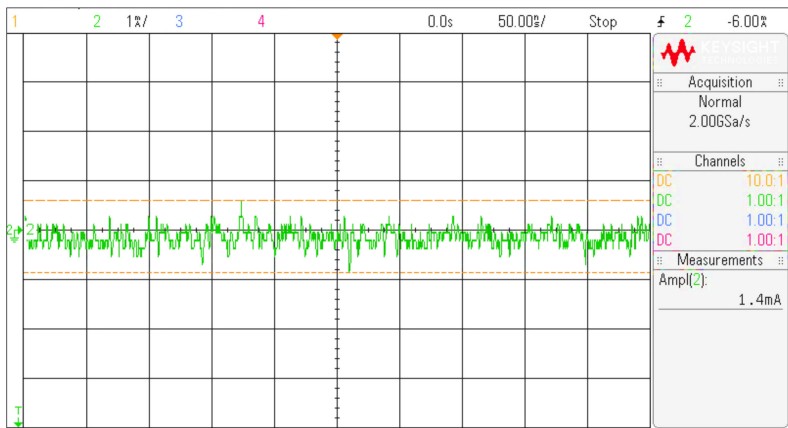

**Figure 20.** Input current of the piezoelectric vibration transducer (PVT).

**Table 6.** Comparison between simulation and experimental outputs.

| Frequency | $V_{rms}$ (Input) | Simulation Result | Experimental Results | Optimization Algorithm |
|---|---|---|---|---|
| 60 Hz | 300 mV | 7 V | 6.94 V | LSA-PI |
| | | | 6.94 V | BSA-PI |
| | | | 6.96 V | PSO-PI |

**Table 7.** Estimated and measured parameter values of the prototype.

| Parameter | Measured Values |
|---|---|
| Input voltage ($V_{in}$) | 300 mV |
| Output voltage ($V_{out}$) | 7 V |
| Input current ($I_{in}$) | 1.4 mA |
| Input power ($P_{in}$) | 0.28 mW |
| Switching frequency ($f_s$) | 10 kHz |
| Diode threshold voltage ($V_F$) | 0.169 |
| Duty ($U$) | 0.95 |
| Resistance of the MOSFET ($r_{DS}$) | 7.92 |
| Inductor resistance, ($r_L$) | 6.87 |
| Diode forward resistance ($R_F$) | 0.171 |

Table 8 presents the impact of the PEHS using the LSA-PI voltage controller, which was verified with current and another conventional work based on voltage, switching frequency, converter proficiency and regulator methods. Besides, the analysis is also suitable for the overall system capability, number of elements required and power loss. Furthermore, the comparison illustrates the advantages and drawbacks of the PEHS converter regulator. Several researchers have developed a boost converter for PEHS, but nobody has utilized optimization methods. In [33], the authors described a step-up converter circuit using 3 MHz switching frequency, to raise the voltage outcome 1.2 V DC for an input of 0.12 V AC. The study reported in [44] presented a boost converter for micro-energy harvesting via a 170 kHz switching frequency to rise the voltage output to 3.3 V DC for a 0.25 V–0.4 V AC input. In [35] the authors developed a step-up converter to rise the voltage output for an energy harvesting system, and the model produces a 4.1 V–5 V DC with an input of 4 mV AC. A drawback of both works was that the authors did not report the converter efficiency. The authors in [36] suggested a flyback converter to raise the voltage output for PEHS without utilizing any optimization method. In [36] the model increased the voltage to 5 V DC with an input of 2.5 V AC. The shortcoming of this study was that the efficiency was not reported and also the work was not validated at the experimental level. The work in [45] addressed a PEHS converter utilizing the BSA optimization technique to improve the voltage output. The proposed system produced 6.06 V DC from an input of 0.3 V AC. The drawbacks of this study were that the efficiency of the voltage output is poor and it did

not mention the overall system efficiency. The study in [46] addressed an LSA technique for enhancing PEHS. However, a limitation of this system was that did not mention the overall efficiency of this system with loss calculation during hardware implementation. Hence, in this work, a LSA-PI voltage controller is used to boost the output voltage for the PEHS converter. The outcome of the LSA-PI voltage controller in the PEHS converter showed a huge enhancement; in comparison with different methods offer better constancy, increased maximum voltage and a faster response based on rise time and settling time, with variable loads, in both the simulation and hardware outputs. The experimental results show that the voltage output is 6.94 V DC for an input of 300 mV AC with a 10 kHz switching frequency.

**Table 8.** Comparison of results between the present and conventional works.

| References | [44] | [45] | [46] | [47] | [48] | [49] | This Study |
|---|---|---|---|---|---|---|---|
| Algorithm | N/A | BSA | LSA | N/A | N/A | NA | LSA-PI |
| $V_{in}$ (Input voltage) | 0.12 V | 0.3 V | 0.25 V | 0.25 V–0.4 V | 40 mV | 2.5 V | 300 mV |
| $V_o$ (Output voltage) | 1.2 V | 6.06 V | 7.05 V | 3.3 V | 4.1 V–5 V | 5 V | 6.94 V |
| Switching Frequency | 3 MHz | 10 kHz | 10 kHz | 170 kHz | 100 kHz | 50.39 kHz | 10 kHz |
| Load | 10 kΩ | Resistive load | Resistive load | 33 kΩ | Not Reported | Not Reported | 2 MΩ, 100 kΩ, 470 Ω, 50 Ω |
| Efficiency | 35% | Not Reported | Not Reported | 70% | Not Reported | Not Reported | 85% |
| Application | Energy harvesting | Energy Harvesting | Energy Harvesting | Energy harvesting | Low power | Energy Harvesting | Micro Devices for Energy Harvesting |

## 7. Conclusions

This study proposed the implementation of a signal-phase PEHS converter prototype utilizing the dSPACE DS1104 controller board based on a hybrid optimization method. This work simulated the system in the MATLAB/Simulink background and experimentally verified the results. The proposed hybrid optimization method was utilized to aid in eliminating the conventional manual tuning technique for seeking suitable values of $K_p$ and $K_i$. To find a solution for the optimum fitness issues in MAE, LSA was utilized, and the simulation and hardware output of this technique were compared with the outputs of the BSA and PSO to validate the results. The obtained output fairly represents that the LSA-PI voltage controller works better than BSA-PI and PSO-PI voltage controllers based on rising time, settling time, constancy, rise height voltage, faster feedback and converter efficiency. Lastly, this prototype efficiently boosts a 300 mV input, with 60 Hz AC to 6.94 V DC. The output voltage is clearly controlled at 6.94 V over a closed-loop utilizing a LSA-PI voltage controller that is appropriate for low power applications. The overall circuit efficiency is about 85%, based on the simulation and the experimental results. This has shown the huge improvement in this study of PEHS using optimization technique compares to the other related work.

**Author Contributions:** Data curation, M.R.S.; funding acquisition, A.H., R.M., and M.H.M.S.; methodology, M.R.S.; resources, R.M., M.H.M.S. and A.M.; software, M.R.S.; supervision, R.M. and A.M.; writing—original draft preparation, M.R.S.; writing—review and editing, M.R.S., M.T. and A.H. All authors have read and agreed to the published version of the manuscript.

**Funding:** This research was funded by Universiti Kebangsaan, Malaysia under Grant Code DIP-2018-020.

**Conflicts of Interest:** The authors declare no conflict of interest.

## Appendix A

| Algorithm A1. | Pseudo Code for LSA Optimization |
| --- | --- |
| | **Setting the parameters** |
| 1 | **Input: I, P, M, i=1 process time** |
| | **Output: Error minimizer** |
| 2 | **for** d = 1:P |
| 3 | Dpoint(d,1)=rand*(upper_kp-lower_kp)+lower_kp; |
| 4 | Dpoint(d,2)=and*(upper_ki-lower_ki)+lower_ki; |
| 5 | kp=Dpoint(d,1); |
| 6 | ki=Dpoint(d,2); |
| 7 | No. of sample a=size(error) |
| 8 | ObjFun(MAE) =sum(abs(error))/a(1); |
| 9 | Evaluation(d)= ObjFun(MAE); |
| 10 | **End** |
| 10 | ch_time = 0; |
| 11 | max_ch_time = 5; |
| 12 | fit_old = 10^10*(ones(1,P)); |
| 13 | direct = sign(unifrnd(-1,1,1,M)); |
| 14 | **for** i = 1:I |
| 15 | Evaluation; |
| 16 | ch_time = ch_time+1; |
| 17 | **if** ch_time>=max |
| 18 | [Ms ds]=sort(Evaluation, 'ascend'); |
| 19 | Dpoint(ds(d),:) = Dpoint(ds(1),:); |
| 20 | Evaluation(ds(d)) = Evaluation(ds(1)); |
| 21 | ch_time = 0; |
| 22 | **end** |
| 23 | [Ms ds]=sort(Evaluation, 'ascend'); |
| 24 | best = Evaluation(ds(1)); |
| 25 | worst = Evaluation(ds(d)); |
| 26 | Energy = 2.05 - 2*exp(-5*(T-t)/T); |
| 27 | **for** i = 1:P |
| 28 | dist=Dpoint(i,:)- Dpoint(ds(1),:); |
| 29 | **for** d = 1:M |
| 30 | **if** Dpoint(i,:)==Dpoint(ds(1),:) |
| 31 | Dpoint_temp(d)=Dpoint(i,d)+direct(d)*abs (normrnd(0,Energy)); |
| 32 | **else** |
| 33 | **if** dist(d)<0 |
| 34 | Dpoint_temp(d) = Dpoint(i,d)+exprnd(abs(dist(d))); |
| 35 | **Else** |
| 36 | Dpoint_temp(d) = Dpoint(i,d)-exprnd(dist(d)); |
| 37 | **End** |
| 38 | **End** |
| 39 | **End** |
| 40 | **if** (Dpoint_temp(1)>upper_kp)\|\|(Dpoint_temp(1)<lower_kp) |
| 41 | Dpoint_temp(1)=rand*(upper_kp-lower_kp)+lower_kp; |
| 42 | **end** |
| 43 | **if** (Dpoint_temp(2)>upper_ki) \|\| (Dpoint_temp(2)<lower_ki) |
| 44 | Dpoint_temp(2)=rand*(upper_ki-lower_ki)+lower_ki; |
| 45 | **end** |
| 46 | kp=Dpoint_temp(1); |
| 47 | ki=Dpoint_temp(2); |
| 48 | fv= ObjFun(MAE); |
| 49 | **if** fv < Evaluation(i) |
| 50 | Dpoint(i,:) = Dpoint_temp; |
| 51 | Evaluation(i) = fv; |
| 52 | **end** |
| 53 | **end** |
| 54 | **if** rand < 0.01 |
| 55 | **for** d = 1:M |
| 56 | Dpoint_fock(d,1) = upper_kp+lower_kp-Dpoint_temp(1); |
| 57 | Dpoint_fock(d,2) = upper_ki+lower_ki-Dpoint_temp(2); |
| 58 | **End** |
| 59 | kp=Dpoint_fock(d,1); |

| Algorithm A1. | Pseudo Code for LSA Optimization |
| --- | --- |
| 60 | ki=Dpoint_fock(d,2); |
| 61 | a=size(error); |
| 62 | ObjFun(MAE)=sum(abs(error))/a(1); |
| 63 | fock_fit= ObjFun(MAE); |
| 64 | **if** fock_fit < Evaluation(i) |
| 65 | Dpoint(i,:) = Dpoint_fock; |
| 66 | Evaluation(i) = fock_fit; |
| 67 | **end** |
| 68 | **end** |
| 69 | Dpoint; |
| 70 | Evaluation; |
| 71 | Fitness(t) = min(Evaluation); |
| 72 | **end** |

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
