# Peer review of "A Hybrid Optimization Approach for the Enhancement of Efficiency of a Piezoelectric Energy Harvesting System"

_electronics, doi:10.3390/electronics10010075_

Round 1
Reviewer 1 Report
- The detailed information of piezoelectric energy harvester should be given, such as structure, dimension.
- In Table 9, why did you use micro devices in your study? Not for energy harvesters?
- Some graphs are not clear, please make redrawing.
- Some new published papers about piezoelectric devices should be introduced. Is there different from your device? Such as, Advanced Functional Materials, 2020, 2000477; International Journal of Mechanical Sciences Volume 189, 1 January 2021, 106003; Nano Energy 76, 104966 (2020); Sensors 2020, 20(12), 3512; IEEE Transactions on Industrial Electronics, 2020, DOI 10.1109/TIE.2020.2978727; “Piezoelectric Energy-Harvesting Interface Using Split-Phase Flipping-Capacitor Rectifier With Capacitor Reuse for Input Power Adaptation, 10.1109/JSSC.2020.2989873.

Reviewer 2 Report
In this manuscript, authors try to improve the piezoelectric energy harvesting efficiency by optimizing the design of converter and controller. Developing a new technique to enhance the piezoelectric energy harvesting efficiency for not only a energy generation mechanism in materials but also a output controlling circuit is very important. In that sense, the contents of this manuscript is important enough to be considered a publication in the Electronics. However, authors should consider following comments before the publication.
1) It would be better if authors could show a final product of PEHS which integrated a piezoelectric energy haversting, converter, and controller units. It seems that if the PEHS includes all necessary units, the size of the PEHS devices may too big for a real application.
2) As authors know, in addition to use a piezoelectricity, there existing more promising vibrational energy harvesting method, triboelectricity. It seems that the optimization method proposed in this manuscript can be also used in a triboelectric energy harvesting system. If so, it would be better change the title of this manuscript.
3) Technical comments
- Appropriate references should be noted for models and theories which already known.
- There exists unnecessary figure in scientific papers such as Fig. 5.
- Table 2 should be moved to Appendix.
- It would be better if the parameters shown in Fig. 13 could be shown in a Table.
- In stead of showing monitor screen pictures, data should be shown as a graph. Figs. 11, 12, 16, 17, 18, 19, and 20 should be replotted.
Reviewer 3 Report
The work, titled “A Hybrid Optimization Approach for the Enhancement of Efficiency of Piezoelectric Energy Harvesting System”, reports on the analysis of an optimization approach for the enhancement of performances of PEHS voltage generation. The proposed structure of controller system was studied by MATLAB/Simulink platform simulation and experimentally tested with a piezoelectric prototype through a dSPACE DS1104 board and an interesting and complex set up. Results show the developed method is able to improve the conversion efficiency up to 85%.
The review found this work quite interesting and original. However, major revisions are needed before being further considered for publication in Electronics. Please, address the following comments:
- Paragraph “3. Proposed Hybrid Optimization Approach PIVC for PEHS Converter” explain the proposed hybrid optimization approach. The reviewer found this paragraph not well explained and suggests some efforts to make it clear to the readers. Especially, sentences from line 201 to 206 makes confused and demanding the understanding.
- In the paragraph “3.2 LSA Theory”, a detailed description of LSA method was included. The reviewer found this description excessively detailed and it makes the reader losing the thread of the argument. The reviewer suggests reviewing this section.
- Table 2 described in the paragraph “3.3 LSA to achieve the ideal PEHS converter” reports about the pseudo code for the LSA optimization. For the sake of manuscript clarity, the reviewer suggests to include this code in supplementary information section.
- Figure 16 in paragraph 6.4 is not clear to the reviewer. The same labels, used to specify the inset zooms, make difficult to distinguish between settling time and input at 300mV. A new arrangement of the figure, and a more detailed description of the caption, could help the reader in a better understanding.
- More than moderate English changes are required all around the text in the manuscript.
Round 2
Reviewer 3 Report
No more revisions are requested. The reviewer recommend to accept the paper in the present form for publication.